# Do Nitrosative Stress Molecules Hold Promise as Biomarkers for Multiple Sclerosis?

**DOI:** 10.3390/ijms26073412

**Published:** 2025-04-05

**Authors:** Moritz Förster, Saskia Räuber, Philipp Albrecht, Lars Wojtecki, Sven G. Meuth, David Kremer

**Affiliations:** 1Department of Neurology, Kliniken Maria Hilf GmbH, Academic Teaching Hospital of the RWTH Aachen University Hospital, 41063 Mönchengladbach, Germany; moritz.foerster@mariahilf.de (M.F.); philipp.albrecht@mariahilf.de (P.A.); 2Department of Neurology, Medical Faculty and University Hospital Düsseldorf, Heinrich Heine University Düsseldorf, 40225 Düsseldorf, Germany; saskiajanina.raeuber@med.uni-duesseldorf.de (S.R.); lars.wojtecki@artemed.de (L.W.); svenguenther.meuth@med.uni-duesseldorf.de (S.G.M.); 3Department of Neurology and Neurorehabilitation, Hospital Zum Heiligen Geist, Academic Teaching Hospital of the Heinrich Heine University Düsseldorf, 47906 Kempen, Germany; 4Institute of Clinical Neuroscience and Medical Psychology, Medical Faculty and University Hospital Düsseldorf, Heinrich Heine University Düsseldorf, 40225 Düsseldorf, Germany

**Keywords:** biomarkers, nitrosative stress molecules, neurofilament light chain, glial fibrillary acidic protein, relapsing remitting multiple sclerosis, primary progressive multiple sclerosis

## Abstract

Multiple sclerosis (MS), an auto-immune disease of the central nervous system (CNS) with inflammatory and neurodegenerative properties, remains an insufficiently understood disease despite more than 150 years of research. In contrast to diseases from other medical fields such as, for instance, oncology, a description of its clinical and non-clinical features based on readouts such as biomarkers is still in its infancy. While, in this regard, neurofilament light chain (NfL) seems to be a promising new tool, the significant intra- and interindividual variation of this serological marker somewhat limits its widespread applicability in everyday clinical reality. This has sparked novel studies in which glial fibrillary acidic protein (GFAP) was proposed as an on-top marker serving to improve overall specificity. In this context, it was found that MS disease progression was significantly more often associated with increased levels of both NfL and GFAP compared to increased NfL levels alone. This highlights the complexity of the disease while also emphasizing the potential benefits of introducing additional markers to enhance current options. We propose that nitrosative stress markers, such as nitrate, nitrite, and nitrotyrosine (3NT), could serve this purpose effectively.

## 1. Introductions

In recent years, nitric oxide (NO) and its metabolites nitrite and nitrate (NOx) have gained increasing attention with regard to multiple sclerosis (MS). For instance, studies have linked atmospheric NOx exposure to an increased risk of developing MS in a dose-dependent manner [1]. This risk is further increased by the presence of the HLA-DRB1*15:01 allele, suggesting a synergistic interaction between genetic predisposition and environmental factors. Additionally, the gut microbiome, which both influences and is influenced by nitrosative species, is being explored for its role in MS pathogenesis, although further research is needed to fully understand these relationships [2,3,4]. Our theory, elucidated here, is that nitrosative stress markers may be a promising addition to the toolbox of MS biomarkers. This rests on our previous results, where we analyzed data found in the literature but also conducted a retrospective analysis of samples from our own patient cohorts [5]. In the afore-mentioned meta-analysis of 22 studies, we found that MS is associated with higher nitrite/nitrate (NOx) levels in the cerebrospinal fluid (CSF) compared to patients with non-inflammatory other neurological diseases (NIOND). Given the significant heterogeneity among existing studies—such as the lack of distinction between clinical MS subtypes, differences in pre-treated versus treatment-naïve patients, and variability in inclusion criteria—we conducted our own study to analyze serum and CSF NOx levels in clinically well-defined cohorts of treatment-naïve MS patients compared to those with somatic symptom disorder. Our findings revealed that RRMS and PPMS patients had higher serum NOx levels than controls. Notably, this difference remained significant within the subgroup of MRZ-negative RRMS patients and double negative (oligoclonal band (ocb)- and MRZ-negative) MS patients, which often pose a significant challenge to the diagnosing clinician [6]. Due to the complexity of MS, it seems self-evident that numerous molecules must be involved in its pathogenesis and, therefore, at least theoretically, many biomarkers could be used both to confirm the diagnosis and to assess disease activity and progression. This review now investigates the relevance, similarities and differences between nitrosative stress molecules and neurofilament light chain (NfL)/glial fibrillary acidic protein (GFAP), which have recently been the subject of intensive research (Table 1).

## 2. Nitric Oxide (NO) and Nitrosative Stress Molecules (NOx) in the Central Nervous System

Nitric oxide (NO) is synthesized from L-arginine via nitric oxide synthase (NOS), which exists in three isoforms: endothelial (eNOS), neuronal (nNOS), and inducible NOS (iNOS). These isoforms are responsible for the endogenous NO production in different tissues, each serving distinct physiological functions. In general, NO plays a pivotal role in maintaining homeostasis across many organ systems, with diverse and integral physiological functions. Its involvement in vasodilation and blood pressure regulation is crucial for cardiovascular health, while its neuroprotective and neurotransmitter functions are vital for normal brain activity [7,8]. In this respect, nNOS and iNOS play a particularly important role. Under physiological conditions, nNOS produces NO, which facilitates, for instance, communication between neurons. In contrast to classical neurotransmitters, NO is able to diffuse freely across cell membranes, thereby exerting an influence on neighboring cells without the necessity of synaptic vesicles. This makes it essential for processes such as synaptic plasticity, which is vital for learning and memory [9,10].

On the other hand, the capacity of NO to mediate neuroprotection is a particularly intriguing aspect in the context of neurodegenerative diseases. In physiological concentrations, NO regulates neuronal survival and protects against oxidative damage by the neutralization of reactive oxygen species (ROS; [11,12]). However, excessive NO production, particularly by iNOS in immune cells, can generate nitrosative stress molecules that lead to neurotoxicity, playing a role in the development of pathological conditions such as stroke and multiple sclerosis [13,14,15]. Nitrosative stress comprises reactive nitrogen species (RNS) including NO, NO_3_^-^, and NO_4_-. INOS is expressed by immune cells such as macrophages and microglia in response to pro-inflammatory cytokines. The NO generated by iNOS plays a dual role in immune responses: it acts as a defense mechanism by eliminating pathogens through the production of RNS but also contributes to demyelination and axonal damage when produced in excess [6] The reaction of nitric oxide (NO) with superoxide (O_2_^−^) results in the formation of peroxynitrite (ONOO−), a toxic metabolite that has the potential to disrupt a wide range of biological molecules. Peroxynitrite can undergo a transition into peroxynitrous acid (ONOOH) and both molecules have the potential to cause direct molecular damage. ONOOH undergoes further degradation into reactive radicals, including one-electron hydroxyl (OH·) and nitrogen dioxide (NO₂), contributing to oxidative stress by oxidizing lipids and nitrating proteins. This is evidenced by the production of 3-nitrotyrosine (3-NT), the “footprint” of nitrosative stress in the brain parenchyma. Furthermore, ONOO− can interact with carbon dioxide (CO₂) to form nitrogen dioxide (NO₂) and carbonate (CO₃·−) radicals, which then oxidize and nitrate proteins and DNA, thereby intensifying cellular damage. Furthermore, elevated NO levels have been demonstrated to impair key cellular processes, including glycolysis and oxidative phosphorylation, by inducing the S-nitrosylation of proteins, which can lead to mitochondrial dysfunction. In addition, in the central nervous system (CNS), NO plays an important role in vasodilation and contributes to the impairment of the blood–brain barrier (BBB), allowing larger molecules and leukocytes to infiltrate the CNS [16].

The growing interest in NO and its metabolites as potential biomarkers in MS underscores the importance of assessing nitrosative stress through serum and CSF measurements of nitrite and nitrate. There are various methods for measuring NO and its stable metabolites [5,17]. Each method has distinct advantages and limitations. The Griess reaction is widely used due to its simplicity and cost-effectiveness, though it only measures nitrite directly and requires enzymatic conversion of nitrate to nitrite for total NOx assessment, which may introduce variability. Chemiluminescence detection offers high sensitivity and the ability to detect low NO concentrations but requires specialized equipment and is susceptible to sample handling conditions. High-performance liquid chromatography (HPLC) with electrochemical detection provides precise quantification of nitrite and nitrate levels but is more expensive and time-consuming, requiring technical expertise. Enzyme-linked immunosorbent assays (ELISA) for NO metabolites are highly specific and suitable for large-scale clinical studies, yet they can be influenced by interfering substances in biological fluids. Mass-spectrometry-based techniques are highly accurate and allow for comprehensive metabolic profiling. However, they demand specialized laboratory infrastructure and expertise, which makes them less accessible for routine clinical use.

Several studies have shown that MS patients have elevated levels of NO and its derivatives in their serum and CSF, especially during relapses [18]. This, in turn, suggests that nitrosative stress is associated with inflammatory processes in MS and reflects disease activity. In addition, we found that elevated NOx levels in the CSF correlate with poorer prognosis and higher scores on the Expanded Disability Status Scale (EDSS; [6]). Whether this is the consequence of an endogenous process or the result of an external environmental factor remains insufficiently clarified at present. However, it has recently been demonstrated that the mean concentration of nitrogen oxides (NO and NO_2_) in the atmosphere is associated with an increased risk of developing MS in a dose-dependent manner. Furthermore, the presence of the HLA-DRB1*15:01 allele, which is linked to an elevated susceptibility to MS, was found to interact synergistically with high NOx exposure. Interestingly, even though NO is easily absorbed via inhalation, leading to an increase in NOx in human blood [19], the primary source of NOx is through water or dietary intake [20]. Regarding food intake, it is also important to note that recent debates have focused on the role of the gut microbiome in MS development and modulation. Nitrosative species have been demonstrated to both modulate and influence the microbiome and, depending on its composition, can also be produced by the microbiome itself [2,3,4]. Finally, NOx could facilitate diagnostic delineation of optic neuritis, as elevated NOx levels have been associated with inflammatory optic nerve damage [21,22]. These findings are particularly relevant in light of the forthcoming 2024 revisions to the McDonald criteria, which place greater emphasis on early diagnosis of MS.

## 3. Neurofilament Light Chain (NfL) and Its Role in MS

Neurofilament light chain (NfL), a neuron-specific cytoskeletal protein, is a promising biomarker for axonal damage and neurodegeneration that offers relevant insights into the pathology underlying MS [23]. NfL is predominantly located within axons, where it plays a crucial role in maintaining structural stability and function. During axonal damage, it is released into the CSF and subsequently leaks into the serum where it can be detected and quantified. In turn, this provides the opportunity to gauge the degree of axonal damage in the brain and spinal cord. While nitrosative stress markers, such as nitric oxide (NO) and its derivatives, reflect inflammatory and oxidative processes, NfL provides a direct measure of physical axonal degradation, rendering it as a more specific marker for neurodegeneration [24,25,26]. In MS, NfL levels are consistently elevated in both the CSF and serum during active disease phases, i.e., relapses, but also in the progressive stages of the disease [27,28]. On a histopathological level, such elevations are derived from axonal injury caused by inflammatory demyelination, a hallmark of MS pathology. Studies have demonstrated that NfL levels are higher in patients with progressive MS and those with severe disability, particularly during relapses or periods of heightened disease activity [23]. Consequently, NfL has emerged as a well-suited tool for monitoring disease progression and evaluating ongoing neurodegenerative processes. Furthermore, NfL levels have been shown to predict long-term outcomes, including the risk of future relapses, disability progression, and brain volume loss, underscoring its prognostic value [29]. A particular strength of NfL as a biomarker is its capacity to detect subclinical disease activity that may not be apparent via conventional clinical assessments or imaging techniques [30]. Accordingly, elevated NfL levels have been found to indicate ongoing axonal damage in the absence of visible lesions on MRI, providing a more sensitive measure of disease activity [31]. This is further complemented by its correlation with clinical measures such as the EDSS and brain atrophy [29,32]. While initially measured in the CSF, the quantification of serum NfL (sNfL) has enhanced its clinical relevance, as it provides a more accessible and less invasive approach facilitating repeated assessments over time. Advances in assay technology, particularly the Single Molecule Array (Simoa) platform, have enabled the reliable detection of sNfL, making it a practical option for routine clinical use [33].

Despite these strengths, NfL has also limitations. One such major challenge is its lack of specificity, as elevated NfL levels can also result from other neurological conditions such as traumatic brain injury (TBI), Alzheimer’s disease (AD), and amyotrophic lateral sclerosis (ALS) [34]. This underscores the absolute necessity for an interpretation of NfL levels in conjunction with clinical and imaging findings in order to avoid misattribution to MS. Additionally, confounding factors such as age, body mass index (BMI), and renal function can influence NfL levels, emphasizing the need for standardization and reference ranges in clinical practice. As a result, NfL is most effectively used as part of a comprehensive diagnostic workup rather than a standalone biomarker. As assay standardization improves and normative data increase, NfL has the potential to become a fundamental element in guiding treatment decisions, monitoring therapeutic efficacy, and identifying patients at risk of rapid disease progression. Furthermore, its role in evaluating the efficacy of emerging therapies, particularly neuroprotective agents, positions NfL to be a critical biomarker for advancing MS research and treatment. In conclusion, NfL represents a significant advancement in the biomarker landscape for MS. While its limitations must be addressed through contextual interpretation and standardization, its potential to transform MS care and research is substantial.

## 4. Glial Fibrillary Acidic Protein (GFAP) and Its Role in MS

GFAP, a cytoskeletal protein specific to astrocytes, plays a pivotal role in maintaining the integrity of the CNS and in the regulation of inflammatory CNS processes. It is a key biomarker of astrocytic reactivity, reflecting the presence of inflammation, scarring, and tissue damage in the brain [35]. While astrocytes typically assist in the preservation of tissue structure and the modulation of immune responses, they often become dysfunctional in MS, contributing to neurodegeneration [36]. Elevated levels of GFAP in the CSF and serum are particularly associated with progressive forms of MS, such as PPMS and secondary progressive MS (SPMS) [37]. These findings indicate that GFAP not only reflects the inflammatory response but also captures the extent of astrocyte-driven neurodegeneration in the CNS [38,39,40]. GFAP is an indicator of chronic and non-acute disease processes as it depicts the glial response and astrocytic activation. Elevated serum GFAP levels have been demonstrated to be correlated with higher EDSS scores, increased MRI lesion load, and progression independent of relapse activity (PIRA), thereby underscoring its significance in tracking MS-related disability progression [37,38]. In advanced stages of MS, where non-relapsing progressive disease is predominant, GFAP appears to be a more reliable indicator of chronic damage in comparison to other biomarkers [41]. It reflects the pathophysiological phenomena of astrogliosis, chronic inflammation, and tissue remodeling that characterize progressive MS phenotypes. These findings underscore the potential of GFAP as a clinical tool for patient stratification [37,38,40].

However, GFAP also has limitations as a biomarker. The high variability in its levels between studies and an inconsistent association with relapses reduce its reliability in reflecting MS activity. Accordingly, some studies report no significant change in GFAP levels during relapses, suggesting that it is primarily relevant for chronic rather than transient inflammatory events [42]. In addition, and similar to NfL, GFAP lacks disease specificity, as elevated levels can also be found in other CNS disorders, complicating its use as a stand-alone diagnostic tool. Moreover, its reliance on sophisticated immunoassay techniques may limit its accessibility in certain clinical settings. Future directions for GFAP as a biomarker in MS should therefore emphasize its integration into multimodal biomarker panels combining parameters of axonal damage (e.g., NfL), neuroinflammation, and astrocytic activation. Research should also aim to standardize GFAP assays, establish reference ranges and validate clinical applicability in different patient populations. A better understanding of the temporal dynamics of GFAP in MS and its interaction with other pathological markers could further enhance its usefulness in monitoring disease progression. However, the uniqueness of GFAP lies in its ability to track chronic astrocytic activation and smoldering inflammation, which are central to the progression of disability in MS. By complementing imaging and other biomarkers, GFAP could therefore play a critical role in the future management of MS [42,43].

## 5. Conclusions

In MS, measuring nitrosative stress molecules, such as nitrate, nitrite, and 3NT, may be a valuable addition to the repertoire of more established biomarkers like NfL and GFAP. While NfL primarily indicates axonal damage and GFAP reflects astrocyte-mediated inflammation and glial activation, nitrosative stress molecules provide a broader view of disease activity. They are particularly relevant for capturing inflammation and the toxic effects of reactive nitrogen species (RNS)—aspects that NfL and GFAP cannot directly measure. This broader scope may prove valuable in cases where established markers fail to offer sufficient diagnostic certainty [6]. However, the utilization of NOx as a biomarker is currently constrained by a number of limitations, as there are several confounding factors, such as hypertension, smoking, hypercholesterolemia, and diabetes mellitus, that may all alter NOx levels. Further complexity is added by environmental factors, including air pollution and specific gut microbiome compositions. Although in our own study NOx demonstrated sufficient specificity for MS when compared to patients with somatic symptom disorder, it remains unclear whether this specificity also extends to closely related conditions, such as inflammatory neurological diseases (IONDs), neuromyelitis optica spectrum disorders (NMOSDs), or acute disseminated encephalomyelitis (ADEM). Additionally, there is a lack of large-scale, homogeneous studies and insufficient information regarding the longitudinal dynamics of NOx concentrations throughout the course of the disease. Furthermore, it remains unclear whether NOx is predominantly produced during the inflammatory or neurodegenerative phases of MS. Moreover, there is currently no established and generally accepted threshold value for defining pathological NOx concentrations. Another challenge is the determination of an appropriate control group for comparison. An adaptation of the methodologies employed for NfL and GFAP could therefore facilitate the standardization of NOx assessment. Accordingly, it would make sense to work with z-scores using standardized cohorts. Notwithstanding these challenges, nitrosative stress molecules offer considerable promise for addressing unanswered questions in MS research. Particularly in the context of RIS and CIS and their transition to clinically definite MS (CDMS), nitrosative stress markers may be integrated into clinical practice as a supplementary tool for assessing disease activity and predicting progression. An earlier diagnosis is crucial for optimizing treatment strategies and improving patient outcomes. In conclusion, nitrosative stress biomarkers hold significant potential for advancing MS research, though further studies are required to validate their clinical utility. Integrating these biomarkers with established tools such as NfL and GFAP could provide a more holistic approach to MS diagnosis, prognosis, and treatment monitoring—particularly in early disease stages, neurocognitive relapses, and tracking disease progression.

## Figures and Tables

**Table 1 ijms-26-03412-t001:** Comparative analysis of NOx/nitrite/nitrate, neurofilament light chain (NfL), and glial fibrillary acidic protein (GFAP) as biomarkers in multiple sclerosis.

Feature	NOx/Nitrite/Nitrate	Neurofilament Light Chain (NfL)	Glial Fibrillary Acidic Protein (GFAP)
Biological role	Reflects nitrosative stress, inflammation, and oxidative damage	Indicates axonal injury and neurodegeneration	Reflects astrocytic reactivity, inflammation, and tissue damage
Disease relevance	Elevated in MS, particularly during relapses; linked to disease activity and poorer prognosis	Elevated during active disease and relapses; correlates with axonal damage and disability	Elevated in progressive forms of MS; reflects chronic damage, glial activation, and neurodegeneration
Sampling	Measured in cerebrospinal fluid (CSF) and serum	Measured in CSF and serum	Measured in CSF and serum
Strengths	Provides insight into inflammation and nitrosative stress mechanisms not captured by other markers	Directly correlates with axonal damage and disease progression; well-established methodology	Strong correlation with chronic disease progression and progressive MS phenotypes
Limitations	Affected by environmental factors, diet and comorbidities; lacks standardization and large-scale validation	High intra- and interindividual variability; elevated levels may also appear in other neurodegenerative diseases	Less sensitive for early disease stages; primarily associated with progressive MS and chronic inflammation
Clinical applications	Potential for early diagnosis and monitoring disease activity; less developed as a standard marker	Monitoring disease progression, activity, and treatment response	Tracking disease progression in progressive MS and stratifying patients by disease phenotype
Integration with others	May complement NfL and GFAP by capturing inflammation and oxidative stress information	Complementary with GFAP for more accurate disease progression assessment	Enhances specificity when used with NfL for progressive MS characterization

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
