# Peer review of "Do Nitrosative Stress Molecules Hold Promise as Biomarkers for Multiple Sclerosis?"

_ijms, 2025, doi:10.3390/ijms26073412_

Round 1

Reviewer 1 Report

Comments and Suggestions for Authors

Dear authors,

The review is focusing on biomarker especially NOx in patients with MS. It has been well known that elevated serum/CSF levels of Neurofilament light chain and GFAP in MS. Although authors demonstrate that MS progression is associated with these molecules and nitrosative stress markers based on their previous study, I could not find novelty from this review. 

There are many reviews that are kind of this. Please emphasise the difference between this review and others.

Comments on the Quality of English Language

Authors have to improve English in this manuscript, since the quality of English language is slightly low. This is not acceptable for publication in this journal.

Author Response

We thank the reviewer for his thoughtful comments. We appreciate his insights and the opportunity to clarify the novelty of our review.

While we acknowledge that there are several reviews on biomarkers in MS, we believe that our work provides a unique perspective by specifically focusing on the role of nitrosative stress markers, particularly NOx, in MS progression. This aspect has not been extensively addressed in previous reviews. Furthermore, our review is, to the best of our knowledge, the first to compare the clinical utility of NOx with well-established biomarkers such as neurofilament light chain (NfL) and glial fibrillary acidic protein (GFAP) in a clinical context. By integrating these perspectives, we aim to highlight the potential complementary role of nitrosative stress markers in MS diagnosis and monitoring, which remains an unmet clinical need. In addition, we have carefully reviewed the manuscript and made improvements to the language for clarity where deemed necessary.

Reviewer 2 Report

Comments and Suggestions for Authors

The authors of the review: “Do nitrosative stress molecules hold promise as biomarkers for multiple sclerosis?”  compare nitrosative stress markers with the NfL and GFAP as a tool in diagnostic process and prognostic factors in patients with multiple sclerosis. The manuscript interestingly addresses the very current problem of searching for biomarkers in MS and is a valuable study for clinicians.

I have a few minor comments:

  • please add the appropriate references for the phrases in the lines: 112-115; 115-117; 128-134; 134-137; 137-140
  • please add in chapter 2 the methods of NOx/nitrite/nitrate assessment (advantages and disadvantages) in serum and CSF in clinical practice

Author Response

We thank the reviewer for his valuable comment.

We have added the relevant references to the manuscript to support the statements in the indicated lines. We also appreciate the suggestion of the reviewer to add a paragraph about methods of NOx/nitrite/nitrate assessment. In response, we have revised chapter 2 to outline the methods used for the assessment of NOx in serum and CSF and briefly discuss advantages and disadvantages (see line 105-121).

Reviewer 3 Report

Comments and Suggestions for Authors

This manuscript examines the role of nitrosative stress molecules, such as nitrate (NO₃⁻), nitrite (NO₂⁻), and 3-nitrotyrosine (3NT), as possible biomarkers for multiple sclerosis (MS). It also discusses how these markers could be combined with already known biomarkers like neurofilament light chain (NfL) and glial fibrillary acidic protein (GFAP). The results suggest that these molecules may improve diagnosis, but some challenges still exist, such as standardization, disease specificity, and clinical validation.

There are several important questions that need to be addressed:

  1. How do you plan to standardize NOx measurements across different laboratories and clinical environments?
  2. How do you consider environmental and metabolic factors that may affect NOx levels and influence your results?
  3. Do you have data showing how NOx levels change over time in MS patients, and how do these changes relate to disease progression or treatment effects?
  4. Why did you choose somatic symptom disorder as a control group, and do you think it is the best choice for comparing NOx levels in MS?
  5. Since GFAP does not always show a clear link with MS relapses, do you think it is a reliable marker for disease activity, or should it be used only for progressive MS?
  6. Since NfL is found in different neurological diseases, how do you make sure it is specific to MS in your study?
  7. Besides validation studies, what steps should be taken to use NOx biomarkers in medical practice, and how would they improve current MS diagnosis methods?

Author Response

We would like to thank the reviewer for the time and effort spent reviewing our manuscript.

We greatly appreciate the thoughtful and constructive comments provided, which have been carefully considered and brought new insights for us. In the following we give a point by point answer:

1) How do you plan to standardize NOx measurements across different laboratories and clinical environments?

We thank the reviewer for raising this important point. As outlined in line 239, we have addressed the standardisation of NOx measurements by suggesting to work with z-scores. This approach helps normalize the data across different laboratories and clinical environments, ensuring that the measurements are comparable. The Z-Score method is one of the most cost-effective and quickest options available for such standardization, providing a reliable way to compare results from diverse settings.

2) How do you consider environmental and metabolic factors that may affect NOx levels and influence your results?

We appreciate the reviewer’s insightful question. As highlighted in line 225, we acknowledge the importance of environmental and metabolic factors in influencing NOx levels. In the manuscript, we briefly mention potential factors, such as hypertension, smoking, hypercholesterolemia, and diabetes mellitus which can impact NOx levels. However, we agree that this is an important area for further discussion. Future studies should record these criteria in order to detect a possible influence on the NfL values and, if necessary, to be able to adjust for this.

3) Do you have data showing how NOx levels change over time in MS patients, and how do these changes relate to disease progression or treatment effects?

Thank you for this very important question. As stated in line 232, we do not currently have data showing how NOx levels change over time in MS patients. We recognize that this is a key aspect of understanding the dynamics of NOx levels in relation to disease progression and treatment response. Although we do not have longitudinal data for this study, we plan to explore this in future research and agree that examining the temporal changes in NOx levels will be crucial for understanding their role in MS.

4) Why did you choose somatic symptom disorder as a control group, and do you think it is the best choice for comparing NOx levels in MS?

We appreciate the reviewer's question about the choice of control group. We chose somatic symptom disorder as a control group because it is a common differential diagnosis in clinical practice when differentiating inflammatory CNS diseases such as MS. This makes it a relevant comparison for assessing NOx levels. We acknowledge that alternative control groups, such as rheumatic diseases with CNS involvement, could have been considered. However, given the size and power of  our previous study, it was not feasible to include these alternative groups. In future studies, we aim to investigate broader control groups to improve the comparability of results.

5) Since GFAP does not always show a clear link with MS relapses, do you think it is a reliable marker for disease activity, or should it be used only for progressive MS?

We agree with the reviewer's comment that GFAP does not always show a clear association with MS relapses. As mentioned in line 195, we have acknowledged this limitation and clarified that GFAP may not be the most reliable marker for MS relapses. However, we believe that GFAP may still be a valuable marker, particularly in progressive MS, where its levels may be more consistently associated with disease activity.

6) Since NfL is found in different neurological diseases, how do you make sure it is specific to MS in your study?

We appreciate the reviewer's important point regarding the specificity of NfL in MS. We agree that NfL is found in several neurological diseases, which raises concerns about its specificity. As highlighted in line 157, we discuss the challenges related to the specificity of NfL in MS. We emphasize that although NfL levels may be elevated in other diseases, its use as a biomarker in MS is particularly relevant when considered in conjunction with MS diagnostic criteria, clinical presentation, MRI findings and differential diagnosis.

7) Besides validation studies, what steps should be taken to use NOx biomarkers in medical practice, and how would they improve current MS diagnosis methods?

We thank the reviewer for pointing this out. We agree that validation studies are essential to establish the utility of NOx biomarkers in clinical practice. One of the first steps is to establish a standard for the use of NOx biomarkers in MS diagnosis. Part of this would be to define a threshold value for NOx, above which a NOx value is considered to be indicative of an MS. After that, NOx biomarkers should be incorporated into routine clinical screening, especially for patients presenting with non-specific neurological symptoms. By incorporating NOx biomarkers into a multiparameter diagnostic approach, clinicians can make more accurate and reliable diagnoses. This reduces the risk of misdiagnosis, particularly in cases of early or atypical MS.This can be particularly useful in ambiguous cases where clinical and imaging findings are inconclusive such as in MRZ-/OKB- patients with MRI abnormalities suggestive of MS.

Round 2

Reviewer 1 Report

Comments and Suggestions for Authors

Dear authors,

Thank you for your revision and comments.

I think now the manuscript will be suitable for publication in the journal.